# Impact of Pb Toxicity on the Freshwater Pearl Mussel, *Lamellidens marginalis*: Growth Metrics, Hemocyto-Immunology, and Histological Alterations in Gill, Kidney, and Muscle Tissue

**DOI:** 10.3390/toxics11060475

**Published:** 2023-05-23

**Authors:** Mohammad Amzad Hossain, Toma Chowdhury, Gourab Chowdhury, Petra Schneider, Monayem Hussain, Bipresh Das, Mohammed Mahbub Iqbal

**Affiliations:** 1Department of Fish Biology and Genetics, Sylhet Agricultural University, Sylhet 3100, Bangladeshmmiqbal5.fbg@sau.ac.bd (M.M.I.); 2Department for Water, Environment, Civil Engineering and Safety, Magdeburg-Stendal University of Applied Sciences, Breitscheidstraße 2, 39114 Magdeburg, Germany

**Keywords:** *Lamellidens marginalis*, LC_50_, specific growth rate, serum lysosomal activity, histopathology

## Abstract

Pb is one of the most extensively used harmful heavy metals in Bangladesh, and its occurrence in waters affects aquatic organisms significantly. The tropical pearl mussel, *Lamellidens marginalis,* was exposed to different concentrations (T_1_ 21.93 mgL^−1^, T_2_ 43.86 mgL^−1^, and T_3_ 87.72 mgL^−1^) of Pb(NO_3_)_2_ and was evaluated against a control C 0 mgL^−1^ of Pb(NO_3_)_2_, followed by a 96 h acute toxicity test. The LC_50_ value was recorded as 219.32 mgL^−1^. The physicochemical parameters were documented regularly for each treatment unit. The values of % SGR, shell weight, soft tissue wet weight, and weight gain remained statistically higher for the control group in comparison with the treatment. No mortality was noted for control units, while a gradually decreased survival rate was recorded for the different treatment groups. Fulton’s condition factor was recorded as highest in the control and lowest in the T_3_ unit, while the condition indices did not vary between the control and treatment groups. The hemocyte was accounted as maximum in the control and T_1_, while minimum in T_2_ and T_3_. The serum lysosomal parameters also followed a similar pattern, and a significantly low level of lysosomal membrane stability, and serum lysosome activity was noted for T_3_ and T_2_ units in comparison to the control group. The histology of the gill, kidney, and muscle was well structured in the control group, while distinct pathologies were observed in the gill, kidney, and muscle tissue of different treatment groups. The quantitative comparison revealed that the intensity of pathological alteration increased as the dosage of Pb increased. The current study, therefore, indicated that intrusion of Pb(NO_3_)_2_ in the living medium significantly alters growth performance and hemocyte counts, and chronic toxicity induces histomorphological abnormalities in vital organs.

## 1. Introduction

Pollution in aquatic environments with a substantial array of pollutants has drawn serious attention globally [1,2]. The discharge of untreated waste with several chemical pollutants and toxicants into water bodies, which is the consequence of modern agricultural practices, rapid urbanization, and industrialization, is responsible for the degradation of aquatic ecological systems [3,4]. Nowadays, heavy metal contamination or pollution of many aquatic habitats is becoming a matter of serious concern [5,6]. Although heavy metals are natural elements, they are likely non-biodegradable in the environment and congest living tissues all along the tropic chain. Heavy metals also tend to geo-accumulate, particularly in fine-grained sediments [7], and most mussels are likely to live on the sediment bed and filter water for food [8].

Lead (Pb) is one of the most toxic heavy metals in nature and is prevalent due to its high industrial application [9]. Like other polluted water bodies globally, Pb pollution in the freshwater bodies of Bangladesh is also becoming an emerging issue, mostly due to the improper management of industrial emissions [6,10]. Since its biological function is still unknown [11,12], the presence of Pb in living tissues signifies the bioaccumulation of this heavy metal from the surrounding environment [10,13]. In aquatic ecosystems, water and food particles are the major pathways through which Pb enters living creatures and accumulates in vital organs [14,15,16]. Several studies have shown that aquatic species pick up this kind of toxic metal following the concentrations present in the neighboring environment [4,17]. The biological equilibrium becomes highly threatened if the concentration of Pb grows beyond the tolerance limit in an aquatic ecosystem, ultimately resulting in the degradation of the entire ecosystem [14,18]. Thus, the higher the heavy metal discharge into the water, the greater the accumulation of heavy metals in the living tissues and sediments [19]. Higher metal concentration causes cellular toxicity and oxidative stress due to the production of reactive oxygen species (ROS) [5,20]. As a result, organisms suffer a lot due to physiological and metabolic dysfunctions, which ultimately destroy ecological health [21]. In addition, this is undoubtedly a sign of danger for the superior consumers, the humans who consume those organisms as food [18].

The freshwater pearl mussel, *Lamellidens marginalis*, is one of the most extensively distributed molluscan animals inhabiting the tropical water of the Indian subcontinent [22]. Like other mollusks, *L. marginalis* have a strong stress tolerance capacity and sensitivity to pollutants, which make them a reliable bio-indicator and sentinel species to identify the pollution level of particular water bodies [23,24]. Moreover, they are considered water cleaners, which can filter a large number of fine particles suspended or dissolved in the water column or bottom sediment, indirectly maintaining the physicochemical properties of aquatic habitats [10,25]. Because of their impressive bioaccumulation capacity, they are being implemented in bio-monitoring programs worldwide to investigate aquatic pollution [23,26].

Nowadays, acute toxicity tests are being implemented as standard tools to measure the effects of toxicants in aquatic habitats [27], where the median lethal concentration (LC_50_) is an extensively applicable and preferable approach in toxicology studies [28]. Besides, previous studies have reported the adverse effects of heavy metals on survival, growth, and the activity of digestive enzymes in molluscans and other aquatic species [14,16,29]. Studies have shown that toxic exposure impairs the membrane stability of lysosomes in hemocytes and causes significant hydrolysis of serum lysosomes [30,31]. Consistently, organisms exposed to higher concentrations of toxicants were found to exhibit not only poor hemolymph status and declined hemocytes in invertebrates [32,33] but also distinguishable histological changes due to heavy metal toxicity in mussels [34,35]. Lead nitrate is one of the extensively tested representatives of Pb in aquatic organisms [36]. Toxicity trials with Pb(NO_3_)_2_ have already been conducted in several species of marine and freshwater mollusks [17,28,37], but very few studies have been conducted on topical freshwater mussels. Therefore, the present study aimed to determine the lethal concentration LC_50_ of Pb in freshwater mussels, and experiments were designed to assess the impact of chronic sublethal lead exposure on growth, hemocyte count, serum lysosome activities, and histological changes in the vital organs.

## 2. Materials and Methods

### 2.1. Collection of Live Animals and Rearing

The present experiment was performed in the wet laboratory of the Department of Fish Biology and Genetics, Sylhet Agricultural University, Sylhet (Bangladesh), from January to February 2022. Mussels were collected from a nearby freshwater body located in Gopalganj, Sylhet, and thereafter transported to the Sylhet Agricultural University in an aerated plastic drum, maintaining all essential safeguards. Then, the mussels were thoroughly rinsed and subjected to 48 h of pre-acclimatization. After that, they were fed dried green algae and thoroughly conditioned in separate aquariums for 14 days before conducting further acute and chronic toxicity assays. Only healthy mussels were considered for both acute and chronic toxicity exposure. A brief overview of activities and laboratory assays in the current research work is outlined in Figure 1.

### 2.2. Acute Toxicity Test and Preparing Concentrations for Chronic Exposure

The acute toxicity test involved assigning 14 dosages of Pb(NO_3_)_2_ (0, 96, 120, 144, 168, 192, 216, 240, 264, 278, 302, 326, and 350 mgL^−1^) based on previous research on mussels to measure the 96 h LC_50_ value. Twenty mussels were accommodated randomly in each treatment unit consisting of 60 L tape water, and ample aeration was ensured by using a centrally located air pump. The rate of mortality was monitored every 24 h, and final values were transformed and plotted in the Probit regression line against the log of toxicant concentration following Finney’s methodology [38]. The LC_50_ value was reported to be 219.32 mgL^−1^, and sublethal doses were assigned as described in Table 1 for further chronic toxicity tests.

Again, glass aquariums filled with 60 L tape water ensuring proper aeration were used for the chronic toxicity test. The initial weight, length, and depth of the mussels are mentioned in Table 4. The mussels were given dried powdered spirulina (Osaka Green-1, Thailand) as food during the experimental period. They were fed at 2% of their body weight three times a day for the first 14 days; afterward, the feeding rate was steadily lowered to 1.5% to maintain uniform water quality for all units. Unused feed and other waste products were siphoned regularly to maintain a suitable environment for the mussels’ existence. Each aquarium was partially refilled with tape water to prevent water loss via evaporation. All tanks were cleaned and washed, and the dosages of Pb were renewed every 14 days.

### 2.3. Sampling and Observation of Hydrological Parameters of Rearing Water

The mussels from all treatment units were randomly sampled to measure the growth parameters every 15 days until the end of the trial. Hydrological parameters, such as temperature, pH, salinity, ammonia (NH_3_), dissolved oxygen (DO), total dissolved solids (TDS), and conductivity, were measured three times during the trial period (1st day, 15th day, and 40th day) using a professional YSI digital multi-Probe meter, Model 58, and a commercial Ammonia Kit (Lifesonic, West Bengal, India).

### 2.4. Tools for Growth Analysis

Specific growth rate SGR (%) = (lnW_2_ − lnW_1_/T_2_−T_1_) ×100, where W_1_ = the initial weight of the mussel at the beginning of the treatment (g), W_2_ = the final weight of the mussel at the end of the treatment (g), and T_2_-T_1_ = duration of the treatment in days. Percentage of weight gain = (W_f_ − W_I_/W_I_) × 100, where W_f_ = mean final weight of the mussel and W_I_ = mean initial weight of the mussel. Percentages of length gain = (L_f_ − L_I_/L_I_) ×100, where L_f_ = mean final length of the mussel and L_I_ = mean initial length of the mussel. The widely used condition index (CI) of the mussels was measured following the description of Uddin et al. [39]. After dissection, the shells were dried in ambient sunlight for 2.5 h before being counted in the calculation. Fulton’s condition factors were calculated by inserting the final shell length and final shell weight into an equation from Htun-Han [40].

### 2.5. Hemocyte Counts, Neutral Red Retention Assay, and Serum Lysosome Activity

At the end of the trial, about 0.5 mL of hemolymph samples were extracted in a centrifuge tube from each animal using a 1 mL syringe and a 0.8 mm needle. The sample hemolymph was then fixed in a similar volume of Alsevier solution (MP Biomedicals, Solon, OH, USA) formulated by putting 3% formaldehyde [41]. The hemocyte concentration was quantified through the hemocytometry approach as described by Munari et al. [42].

Lysosomal membrane stability in hemocytes was evaluated using the Neutral Red Retention Assay (NRRA) [30]. The hemocytes were exposed to 0.01% Neutral Red (Sigma-Aldrich, St. Louis, MO, USA), and the dye retention period for 50% of the lysosome was noted for each of the observations in the microscope. The serum lysosome activity was quantified using the methodology adopted from Chu and Peyre [43] and Capolupo et el. [44]. The method involved the separation of serum from 500 µL of hemolymph by using centrifugation at −4 °C for 10 min. Then the serum was conjugated with *Micrococcus lysodeikticus* (Sigma-Aldrich) in phosphate buffer and the absorbance was spectrophotometrically read at 450 nm in 1 min intervals. The protein content was quantified using bovine serum antibody (BSA) as a standard.

### 2.6. Histology of Gill, Kidney, and Muscle

Previously preserved samples in neutral buffered formalin were washed in tap water and processed according to the description of Slaoui and Fiette [45]. The samples were embedded in paraffin and stained with standard hematoxylin and eosin dye. Finally, the slides were observed in a Primo Star (ZEISS, Oberkochen, Germany) light microscope equipped with an Axiocam Camera (ZEISS, Oberkochen, Germany), and pictures were taken in ZEN core v3.0. At least 10 sample slides were studied for each treatment unit, and pathologies were recorded for further analytical purposes. Histopathology was identified by following the description of McElwain and Bullard [46] and Carella et el. [47], all the slides were observed, and pathology found to be below 5% was marked as absent, 6–25% as weak, 26–50% as moderate (**), and above 50% as severe (***).

### 2.7. Statistical Tools

The data were valued using a one-way analysis of variance at *p* < 0.05 in SPSS v26. Tukey’s HSD post hoc test was used to compare individual means. The hypotheses of normal distribution and homogeneity were verified before conducting the one-way ANOVA. Line graphs and bar diagrams were drawn in Excel, Office 365, using output data from SPSS v26.

## 3. Results

### 3.1. Acute Toxicity Test

The 96 h Pb(NO_3_)_2_ LC_50_ in *L. marginalis* was analyzed using the Probit regression model and found to be 219.32 mgL^−1^. A meta table was made to represent the Pb(NO_3_)_2_ toxicity in different molluscan animals (Table 2). The intensity of Pb(NO_3_)_2_ LC_50_ varied depending on the species of mollusk used in the trial.

### 3.2. Water Quality Parameters in Different Trial Units

Table 3 shows the water parameters that were measured during the trial period. Most of them remained statistically consistent (*p* < 0.05).

### 3.3. Growth Performance, Mortality Rate, and Condition Indices

The overall growth performance is summarized in Table 4. Compared to the control, soft tissue wet weight (g) was reduced in the treatment groups (T3 < T2 < T1) (*p* < 0.05). Again, the lowest values of final shell weight (g) and specific growth rate % were reported in T3, followed by T1 and T2 (*p* < 0.05). Significantly lowest values were noted in T3 for weight gain (−2.49 ± 3.95 g) in contrast to the control unit (8.92 ± 3.52 g) (*p* < 0.05). The lowest mortality was noted in the control group, whereas individuals in the treatment groups experienced a gradual increase in mortality (Figure 2). The lowest survival rate was observed in T3, followed by T2 and T1, in comparison with the control group (Figure 2A) (*p* < 0.05). However, condition indices (CI) among the different treatment and control groups stayed statistically steady, while Fulton’s (F_k_) showed significant differences (Figure 2B).

### 3.4. Hemocytes Count, Lysosomal Membrane Stability (LMS), and Serum Lysosome Activity

Hemocyte counts varied radically between different treatment units, with higher counts in the control and T_1_ groups and lower in T_3_ and T_2_ groups (*p* > 0.05) (Figure 3A). The retention period of neutral red for LMS and serum lysosome activity significantly differed between the control and treatment groups (Figure 3B,C). All the treatment groups exhibited a quick neutral red retention period and decreased value for serum lysosome activity.

### 3.5. Histopathology of Gills, Kidneys, and Muscle

Gill filaments from the control group showed a healthy condition consisting of well-structured lateral cilia, gill epithelium, and hemolymph vessel (Figure 4A), whereas filaments from the treated groups revealed tissue rupture, extended vessel, and cilia damage (Figure 4B–D). Furthermore, congestion in filaments and epithelial deformities were identified in T_1_ and T_2_, respectively (Figure 4B,C), while filaments from T_3_ were affected by the massive expansion of hemolymph vessels, severe epithelial damage, and anomalous filament attachment (Figure 4D). Again, healthy renal tissues were detected by regular epithelial vesicles, brown intracellular granules, connective tissue, and nephridial lumen (Figure 5A). On the contrary, increased granules quantity, necrosis, tissue rupture, and extended lumen were noted in the exposed groups (Figure 5B–D). In addition, epithelial lifting was prominent in T_2_ (Figure 5C), while T_3_ exhibited damage to connective tissue, large vacuoles due to massive tissue rupture, and the presence of enlarged brown intracellular granules (Figure 5D). Normal muscle structure with horizontal and vertical musculature and irregular fibrous tissues were noted in the control group (Figure 6A); in contrast, the treated groups exhibited cell rupture and necrosis as common pathological disorders (Figure 6B–D). Apart from these alterations, massive vacuoles and fibrous tissue damage also appeared in T_2_ and T_3_, respectively (Figure 6C,D). Table 5 summarizes a comparative investigation of histopathological changes in the gills, kidneys, and muscles of *L. marginalis* detected from the present experiment.

## 4. Discussion

LC_50_ is a decisive scientific approach that is widely valid in the study of aquatic toxicology and pollution [5,28,49]. The 96 h LC_50_ of Pb(NO_3_)_2_ obtained from the current study was concomitant with the findings of [48]. The intensity of Pb(NO_3_)_2_ LC_50_ in different studies varied depending on the species, organism size, life stage, and physicochemical properties of the rearing water [17,50]. However, most data suggested features of strong tolerance towards Pb(NO_3_)_2_ toxicity in aquatic mollusks. Water quality parameters are vital in evaluating the toxicity of an aquatic environment [2,51]. Significant fluctuations in the physicochemical features of water may stimulate a species to be more sensitive to toxicants [52]. Hence, the physicochemical properties in the current experiment were kept in the optimum range, following the standard requirements [53].

Toxicology studies have not only signified the adverse effect of heavy metals on the growth performance of mussels, but also speculated the growth rate as one of the sensitive bio-indicators [29,54,55]. According to the present findings, the growth and development of *L. marginalis* were hindered when exposed to Pb(NO_3_)_2_, probably due to unprivileged physiological performance resulting in retarded tissue development and shell formation. The present results were comparable to those of Chandurvelan et al. [14] and Hu et al. [56], while *Perna canaliculus* was tested through exposure to several trace elements, and *Mytilus coruscus* was treated with toxic nanoparticles, respectively, which eventually resulted in poor physiological conditions and growth performance. Reduced growth metrics have also been consistently observed in several fish species owing to lead toxicity [10,18,57]. In contrast, the effect of heavy metals on growth was insignificant when *Lampsilis siliquoidea* was exposed to copper [58]. In the current study, the survival rate was also a significant stress indicator that reflected the severity of lead toxicity. As a result, the highest mortality was reported at the top exposure level [59]. Several previous studies have suggested that toxicity causes significant alterations in hemocyte counts in mussels, and it can be considered a potential biomarker to access environmental health [60,61]. Hemocyte counts significantly reduced when *Mytilus coruscus* was treated with titanium dioxide for 35 days [42], and when *Mytius edulis* was exposed to cadmium for 7 days [62]. Cellular abnormalities and impaired counts were recorded in the zebra mussel *Dreissena polymorpha* in response to cadmium toxicant [63]. The notable reduction in hemocyte count for the high-concentration treatment units of the current study remained aligned with the above findings. Serum lysosome activity is one of the widely used biomarkers in the assessment of toxicity compensation changes in the hematocytes of mussels [42,64]. A trend of the quick neutral red retention period and reduced serum lysosome concentration has been noted for *Mytilus galloprovincialis* exposed to micro- and nanoplastics [31,44].

Heavy metal toxicity causes notable histological changes in the vital organs of invertebrates [35,65,66]. Mussels’ gills are considered to be sensitive to toxicity because of their crucial role in respiration, food absorption, and extended surface area [67,68]. Fahmy and Sayed [33] observed gill filament dilation, epithelial lifting, necrosis, hemocyte congestion, hyperplasia, filament deformities, and cilia damage in bivalve *Coelatura aegyptiaca* exposed to zinc nanoparticles. Similarly, hemocyte infiltration, epithelial damage, necrotic and ruptured cells, and deformed gill filaments were identified by Vasanthi et al. [34]. These pathological alterations were mostly analogous to the present results. Meanwhile, in contrast to the control, several pathological pieces of evidence were obtained from the exposed units in the current experiment. However, the study of Knowles et al. [69] was consistent with the current findings, while expanded renal tubules, enlarged cytoplasmic vacuoles, and slightly expanded renal cells were spotted in the oysters *Crassostrea gigas* due to inferior salinity stress. Pathological signs in renal tissues, such as deformed tubules expanded vacuoles with cytoplasm, tubular necrosis, and squeezed glomerulus, were documented in *Cyprinus carpio* exposed to Pb(NO_3_)_2_ [36]. In the present study, histopathological alterations in foot muscle structure were noticeable, thereby indicating the severity of Pb toxicity. Again, data lacking on invertebrate muscle histopathology hindered the interpretation of the results. Current results were comparable with those of Hossain et al. [49], according to whom degenerated muscle fiber, extended vacuoles, cellular rupture, and necrosis were the prominent pathologies in *Oreochromis niloticus* muscle tissue exposed to chlorpyrifos. Fragmented epithelium and shortened epithelial cells were the major signs of mantle tissue damage in *Mytilus galloprovincialis* when exposed to ZnCl_2_ [35]. Therefore, the current experiment suggested that histology of the gill, kidney, and muscle and serum lysosomal activities might be considered significant tools in investigating metallic toxicity in freshwater pearl mussels.

## 5. Conclusions

Aquatic invertebrates are inevitably impacted by metal contaminants in water, and they act as potential biomarkers for water quality deterioration. Harmful heavy metal contamination in water and bioaccumulation in mussels could lead to public health issues. The results indicated that Pb exposure has a substantial impact on the physical health of *L. marginalis*. The experimental trial showed an increasing impact correlating with increased exposure. Chronic toxicity represented moderate to severe alterations in internal cellular structure. The current study also revealed the vulnerability status of freshwater ecosystems near large cities and industries due to heavy metal pollution. Therefore, necessary measures and policies should be implemented to protect mollusks from harmful heavy metal exposure.

## Figures and Tables

**Figure 1 toxics-11-00475-f001:**
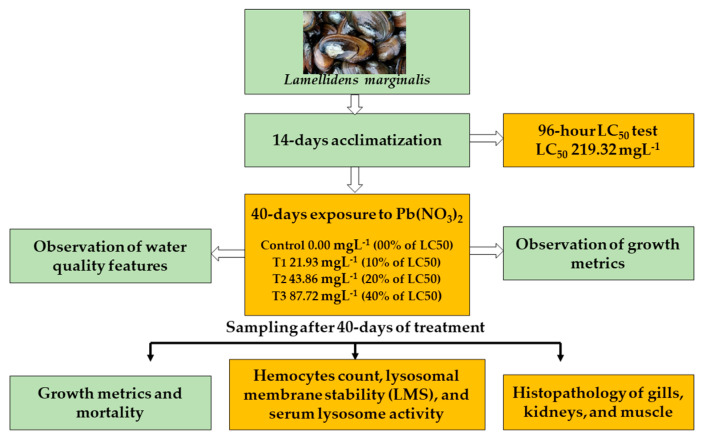
Flow diagram of the overall laboratory protocols and assays.

**Figure 2 toxics-11-00475-f002:**
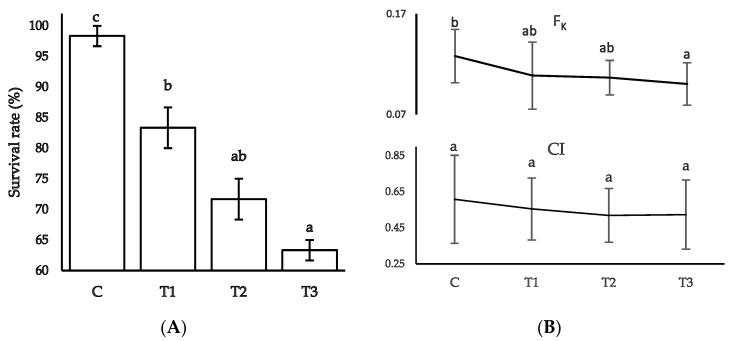
The survival rate and condition indices in the different experimental units. (**A**) Survival rate, (**B**) Fulton’s condition factor F_k_, and condition index, CI). Different superscripts indicate significant differences at *p* < 0.05; values are means ± SE.

**Figure 3 toxics-11-00475-f003:**
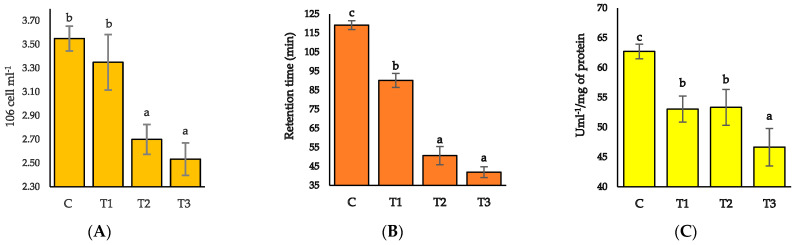
Hemocyte count, lysosomal membrane stability (LMS), and serum lysosome activity in different experimental units following exposure to Pb(NO_3_)_2_. (**A**) Hemocyte count, (**B**) Lysosomal membrane stability (LMS), and (**C**) Serum lysosome activity). Different superscripts indicate significant differences at *p* < 0.05; values are means ± SE.

**Figure 4 toxics-11-00475-f004:**
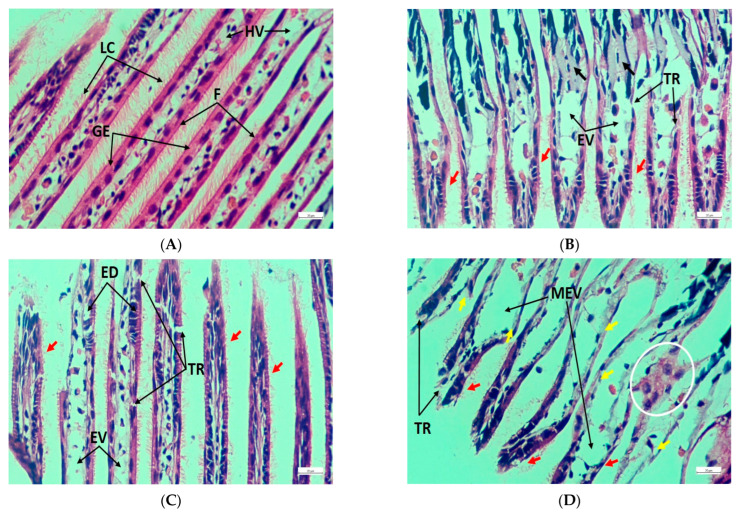
Gill transverse section photomicrograph of *L. marginalis*. (**A**) Control, (**B**) T_1_, (**C**) T_2_, and (**D**) T_3_; LC—Lateral cilia, F—Filaments, GE—Gill epithelium, HV—Hemolymph vessel, TR—Tissue rupture, EV—Extended vessel, MEV—Massive expansion of vessel, ED—Epithelial deformities; Filament congestion—Black arrow, Cilia damage—Red arrow, Epithelial damage—Yellow arrow, Filament attachment, and abnormalities—White circle.

**Figure 5 toxics-11-00475-f005:**
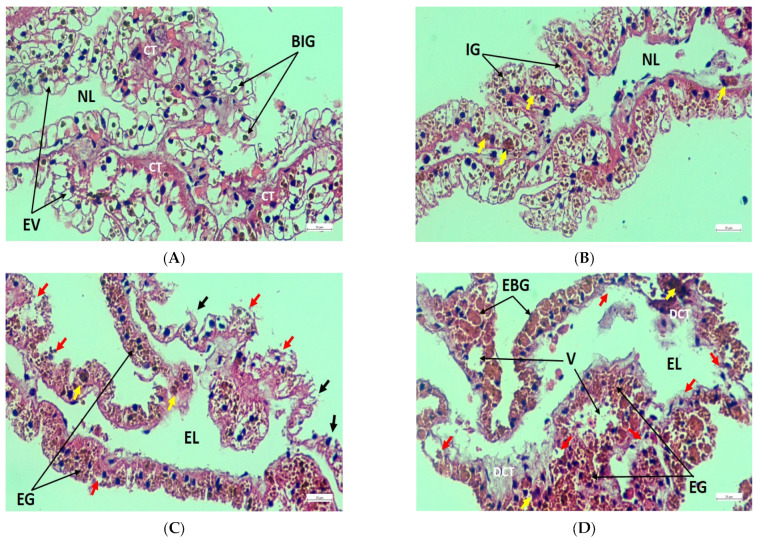
Renal transverse section photomicrograph of *L. marginalis*. (**A**) Control, (**B**) T_1_, (**C**) T_2_, and (**D**) T_3_; EV—Epithelial vesicles, BIG—Brown intracellular granules, CT—Connective tissue, NL—Nephridial lumen, IG—Increased granules, EG—Enormous granules, EL—Extended lumen, DCT—Damage of connective tissue, V—Vacuoles due to massive tissue rupture, EBG—Enlarged brown intracellular granules; Necrosis—Yellow arrow, Tissue rupture—Red arrow, Epithelial lifting—Black arrow.

**Figure 6 toxics-11-00475-f006:**
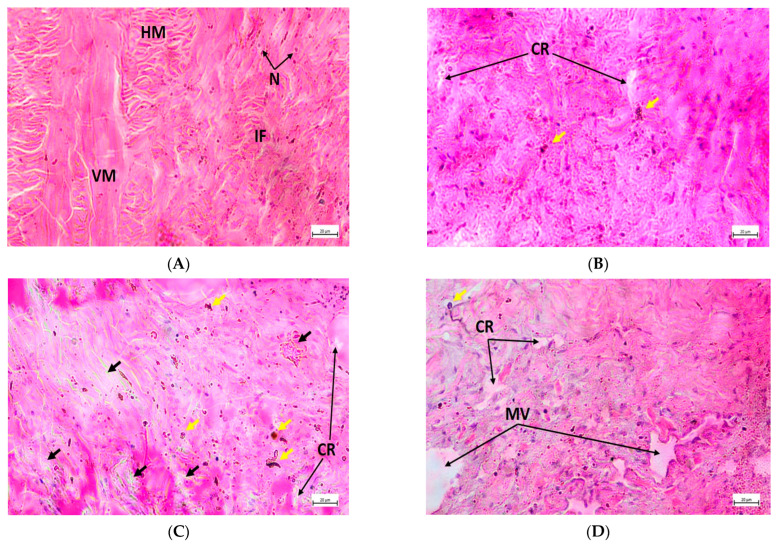
Muscle transverse section photomicrograph of *L. marginalis*. (**A**) Control, (**B**) T_1_, (**C**) T_2_, and (**D**) T_3_; VM—Vertical musculature, HM—Horizontal musculature, IF—Irregular fibrous tissue, N—Nuclei, CR—Cell rupture, MV—Massive vacuoles due to cell rupture; Necrosis—Yellow arrow, Fibrous tissue damage—Black arrow.

**Table 1 toxics-11-00475-t001:** Dose calculation in different treatment groups.

Treatment	Concentration of Pb(NO₃)₂ (mgL^−1^)	Stocking Density (No./Treatment)	Replication
C (Control)	00.00 (00% of LC_50_)	20	3
T_1_	21.93 (10% of LC_50_)	20	3
T_2_	43.86 (20% of LC_50_)	20	3
T_3_	87.72 (40% of LC_50_)	20	3

**Table 2 toxics-11-00475-t002:** 96 h Pb(NO_3_)_2_ LC_50_ in different aquatic mollusks.

	Species	Value of 96 h LC_50_ (mgL^−1^)	References
**Bivalves**	*Lamellidense marginalis*	219.32	Current study
	*L. jenkinsianus*	192.14	[28]
	*Corbicula striatella*	231.01	[48]
**Gastropods**	*Theodoxus niloticus*	18	[5]
	*Melanoides tuberculata*	6.82	[17]
	*Cerithedia cingulate*	15.50	[37]
**Bivalves**	*Modiolus philippinarum*	2.86	

**Table 3 toxics-11-00475-t003:** Water quality parameters in the different treatment groups (values are mean ± SE).

	Treatment	Day 0	Day 15	Day 40
Temperature	C	20.73 ± 0.23	19.56 ± 0.37	20.66 ± 0.50
	T_1_	20.93 ± 0.11	20.23 ± 0.41	20.66 ± 0.20
	T_2_	21.00 ± 0.17	19.70 ± 0.17	20.76 ± 0.15
	T_3_	21.03 ± 0.28	20.13 ± 0.49	20.80 ± 0.10
pH	C	7.96 ± 0.05	8.20 ± 0.10	8.23 ± 0.10
	T_1_	7.83 ± 0.05	8.19 ± 0.10	7.96 ± 0.05
	T_2_	7.70 ± 0.03	8.16 ± 0.05	7.76 ± 0.05
	T_3_	7.60 ± 0.01	8.06 ± 0.05	7.53 ± 0.05
Salinity	C	0.07 ± 0	0.08 ± 0	0.10 ± 0
	T_1_	0.07 ± 0	0.08 ± 0	0.11 ± 0
	T_2_	0.07 ± 0	0.08 ± 0	0.11 ± 0
	T_3_	0.07 ± 0	0.08 ± 0	0.09 ± 0
NH_3_	C	0.05 ± 0	0.07 ± 0	0.07 ± 0
	T_1_	0.01 ± 0	0.02 ± 0	0.02 ± 0
	T_2_	0.02 ± 0	0.02 ± 0	0.02 ± 0
	T_3_	0.02 ± 0	0.01 ± 0	0.02 ± 0
DO	C	7.26 ± 0.49	7.48 ± 0.47	7.39 ± 0.63
	T_1_	7.58 ± 0.29	7.58 ± 0.29	7.60 ± 0.48
	T_2_	7.54 ± 0.075	7.54 ± 0.07	7.43 ± 0.08
	T_3_	7.64 ± 0.104	7.64 ± 0.10	7.77 ± 0.08

**Table 4 toxics-11-00475-t004:** Growth parameters of *L. marginalis* exposed to different concentrations of Pb(NO_3_)_2_.

	C	T_1_	T_2_	T_3_
Shell Length Initial (cm)	7.23 ± 0.16	7.16 ± 0.12	7.26 ± 0.17	7.22 ± 0.20
Shell Weight Initial (g)	41.40 ± 1.8	41.28 ± 2.01	41.06 ± 2.59	41.16 ± 2.52
Shell Depth Initial (cm)	4.06 ± 0.11	4.02 ± 0.09	4.09 ± 0.06	4.01 ± 0.10
Shell Length Final (cm)	7.33 ± 0.09	7.28 ± 0.10	7.31 ± 0.10	7.28 ± 0.13
Shell Weight Final(g)	50.32 ± 2.58 ^b^	40.69 ± 1.58 ^a^	41.41 ± 1.58 ^a^	38.67 ± 2.24 ^a^
Shell Depth Final (cm)	4.32 ± 0.05	4.25 ± 0.05	4.36 ± 0.04	4.36 ± 0.05
Soft Tissue Wet Weight (g)	9.41 ± 0.64 ^b^	7.88 ± 0.26 ^a^	7.85 ± 0.31 ^a^	7.01 ± 0.42 ^a^
Dry Shell Weight (g)	16.45 ± 0.98	15.05 ± 0.85	16.23 ± 1.07	15.09 ± 1.50
Length Gain (cm)	0.10 ± 0.21	0.12 ± 0.13	0.05 ± 0.20	0.06 ± 0.25
Weight Gain (g)	8.92 ± 3.52 ^b^	−0.59 ± 2.65 ^ab^	0.35 ± 3.29 ^ab^	−2.49 ± 3.95 ^a^
Specific Growth Rate %	0.48 ± 0.18 ^b^	−0.01 ± 0.17 ^a^	0.07 ± 0.19 ^a^	−0.13 ± 0.25 ^a^
Average Daily Length Gain	0.002 ± 0.05	0.003 ± 0.003	0.001 ± 0.005	0.001 ± 0.006
Average Daily Weight Gain	0.22 ± 0.08 ^b^	−0.01 ± 0.06 ^ab^	0.01 ± 0.08 ^ab^	−0.06 ± 0.09 ^a^

Notes: The columns with different superscripts indicate significant differences at *p* < 0.05; values are means ± SE.

**Table 5 toxics-11-00475-t005:** Comparative study of histopathological changes from the present trial in different experimental units.

Organ	Abnormality	Control	T_1_	T_2_	T_3_
Gill	Tissue Rupture	---	*	*	**
	Extended Vessel	---	**	**	***
	A Massive Expansion of The Vessel	---	*	---	**
	Epithelial Deformities	---	---	*	*
	Filament Congestion	---	*	---	*
	Cilia Damage	---	*	**	***
	Epithelial Damage	---	---	*	***
	Filament Attachment And Abnormalities	---	---	---	*
Kidney	Enormous Granules	---	*	**	**
	Extended Lumen	---	---	*	**
	Damage to Connective Tissue	---	---	---	*
	Vacuoles Due to Massive Tissue Rupture	---	---	*	**
	Enlarged Brown Intracellular Granules	---	---	*	***
	Necrosis	---	**	**	**
	Tissue Rupture	---	---	**	**
	Epithelial Lifting	---	---	*	---
Muscle	Cell Rupture	---	*	***	**
	Massive Vacuoles Due to Cell Rupture	---	---	**	*
	Necrosis	---	*	*	**
	Fibrous Tissue Damage	---	---	*	***

Notes: Absent (---), below 10% as weak (*), 10–50% as moderate (**), and above 50% as severe (***).

## Data Availability

Raw and analyzed data in the current research are available from the corresponding author based on reasonable request.

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
