# Peer review of "Impact of Pb Toxicity on the Freshwater Pearl Mussel, Lamellidens marginalis: Growth Metrics, Hemocyto-Immunology, and Histological Alterations in Gill, Kidney, and Muscle Tissue"

_toxics, 2023, doi:10.3390/toxics11060475_

Round 1

Reviewer 1 Report (Previous Reviewer 3)

This revised research article was well edited based on reviewers' comments.

Thus this research article may be suitable for the publication of one of research articles in the journal, Toxicity.

Author Response

Dear Editor ,

Thank you. Please check the attachment as authors response letter.

Kind regards.

Reviewer 2 Report (Previous Reviewer 2)

Accept in present form

Author Response

Dear Editor ,

Thank you. Please check the attachment as authors response letter.

Kind regards.

This manuscript is a resubmission of an earlier submission. The following is a list of the peer review reports and author responses from that submission.

Round 1

Reviewer 1 Report

The possible effects of Pb(NO3)2 on the freshwater mussel Lamellidens marginalis are discussed in the text. Growth metrics, hemocytic responses, and the histology of the gills, kidneys, and muscle were evaluated after 40 days of exposure to three doses of Pb(NO3)2. Authors conclude that Pb(NO3)2 exherts toxic effects in mussels and suggest that L. marginalis could be used as sentinel species for future studies. The English in the manuscript needs to be thoroughly revised. There are errors and mistakes  throughout the manuscript.

The abstract is quite ambiguous and nonspecific. The differences between the control and treatment groups are just briefly mentioned by the authors, who do not elaborate on their nature or severity. Vague descriptions.

The experimental procedure needs to be described in more detail. It is unclear whether the Pb(NO3)2 concentration in the water varies over the course of the investigation or remains constant. How frequently the water and Pb doses were renewed is not reported. Also, the authors state in line 111 that water was added to counteract the effects of evaporation. What volume of water was added? How frequently? How might such addition change the level of Pb?

Mussels were collected from the field. Is this environment polluted by Pb? Did you determine the Pb levels in catched mussels?

In line 138 authors mention they used hemocytometry. Can you give further details or a Reference for this procedure?

Line 154: How many slides did you analyse?

Statistical section: Did you check data normal distribution and homogenity? I suggest using only upscript letters in Tables and Figures when statistical differences among groups for a given parameter are determined.

Table 2: This Table should not be included in the Results. I suggest placing it as Supporting Information. Also to consider, that in the Table authors include both Bivalves and Gastropods, freshwater and marine species. I suggest including a column identiying those differences.

Reviewer 2 Report

The paper presents an interesting and important toxicological approach in a key model bioindicator species that should be emphasized specially considering the study area, however there are some english errors and some parts needs clarification while others needs corrections.

Also, Lamellidens marginalis is one of the most widely harvested freshwater bivalve species used for human consumption in India, Nepal and Bangladesh. Hence, it is important to address this issue of risk assessment regarding this species consuption. 

Specific comments are in the file in attach.

Reviewer 3 Report

This research article was well edited based on authors' original research results. However, there were some problems including editing errors. Thus this research article should heve to revise before the publication of one of research articles for international journal(s).

Please refer followings.

[Major reversion]

1) Title

Please use more simple and specific title in order to easy understand.

2) "Heavy metals" ---> "Harmful heavy metals" in all contents.

Gold or silver is one of heavy metals. Do you think that gold is harmful chemical?

3) Keywords

Please use only important terms as "Keywords" at least 5 words.

4) Objective(s)

What is different from that of other researchers' research?

Authors need to consider that why is this research important on the viewpoint of globalization. And authors need to consider the applicability of this research.

5) "4. Discussion and Conclusions"

Please separate "Discussion" part from "4. Discussion and Conclusions" part.

6) Conclusions

I think that these contents are too general looks like just summary of experimental results.

Please revise and add authors' originality, understandings.

Please add the applicability of this research.

Please consider that where and hoe can we apply based on the results of this research?

[Minor reversion]

7) Table 3.

Please use same effective numbers at least 3 or more.

Ex.) EFNs 3: 20.7, 20.9, 21.0, 0.070, 7.26, 100 etc.

8) Table 4.

Please use same effective numbers at least 3 or more.

9) Figure 2.

Please use figure caption and also sub-title for A and B.

10) Figure 3.

Please use figure caption and also sub-title for A, B and C.

11) Figure 4.

Please use figure caption and also sub-title for A, B, C and D.

12) Figure 5.

Please use figure caption and also sub-title for A, B, C and D.

13) Figure 6.

Please use figure caption and also sub-title for A, B, C and D.

14) Page 13, Line 381~382.

Please use the "year" of reference eventhough law(s) or guideline(s).